# Cubesats Allow High Spatiotemporal Estimates of Satellite-Derived Bathymetry

**Dimitris Poursanidis [1],\*** , **Dimosthenis Traganos [2]** , **Nektarios Chrysoulakis [1]** and **Peter Reinartz [3]**

1   Foundation for Research and Technology—Hellas (FORTH), Institute of Applied and Computational Mathematics, N. Plastira 100, Vassilika Vouton, 70013 Heraklion, Greece; zedd2@iacm.forth.gr
2   German Aerospace Center (DLR), Remote Sensing Technology Institute, Rutherfordstraße 2, 12489 Berlin, Germany; Dimosthenis.Traganos@dlr.de
3   German Aerospace Center (DLR), Earth Observation Center (EOC), 82234 Weßling, Germany; peter.reinartz@dlr.de
\*   Correspondence: dpoursanidis@iacm.forth.gr; Tel.: +30-2810391774

**Abstract:** High spatial and temporal resolution satellite remote sensing estimates are the silver bullet for monitoring of coastal marine areas globally. From 2000, when the first commercial satellite platforms appeared, offering high spatial resolution data, the mapping of coastal habitats and the extraction of bathymetric information have been possible at local scales. Since then, several platforms have offered such data, although not at high temporal resolution, making the selection of suitable images challenging, especially in areas with high cloud coverage. PlanetScope CubeSats appear to cover this gap by providing their relevant imagery. The current study is the first that examines the suitability of them for the calculation of the Satellite-derived Bathymetry. The availability of daily data allows the selection of the most qualitatively suitable images within the desired timeframe. The application of an empirical method of spaceborne bathymetry estimation provides promising results, with depth errors that fit to the requirements of the International Hydrographic Organization at the Category Zone of Confidence for the inclusion of these data in navigation maps. While this is a pilot study in a small area, more studies in areas with diverse water types are required for solid conclusions on the requirements and limitations of such approaches in coastal bathymetry estimations.

**Keywords:** satellite-derived bathymetry; hydrography; CubeSats; hypertemporal; zones of confidence; PlanetScope

## 1. Introduction

Bathymetry is the center of several important biogeophysical processes such as primary production and the development of marine forests and seagrass meadows—influenced by the exponential decrease of light with depth. The spatial variation can also define the topographic properties of the studied seascape, e.g., slope, aspect, rugosity, terrain roughness, and the bathymetric position index [1,2]. The importance of bathymetry as a product and its use in nautical charts [3] under several categories (Zones of Confidence—ZOC) is high in areas of maritime navigation. Traditionally, bathymetry has been estimated by the implementation of hydroacoustic tools and methods like the Single-Beam (SBES) and Multi-Beam Echo Sounders (MBES), Airborne Lidar Bathymetry (ALB), and LIDAR devices, installed on vessels following specially designed sailing lines with a specific geometry [4]. These methods, and especially the MBES and ALB, can provide highly accurate information in multiple scales. However, depending on the extent of the project area, they require a large amount of effort

and are costly [5] when compared with newly adopted approaches such as the Satellite-derived bathymetry (SDB).

During the last four decades, many SDB-related studies have emphasized the potential utilization of satellite remote sensing sensors for bathymetric calculations in clear shallow waters in a plethora of spatiotemporal resolutions: From Lyzenga [6,7] in 1978 and 1981 using the first Earth Observation satellite, the optical Landsat Multispectral Scanner (spatial resolution of about 79 m and temporal one of 18 days) to 1983 and the implementation of the first spaceborne synthetic aperture radar satellite, SEASAT (25-m spatial resolution) [8]; and from an inversion of spaceborne altimetry data from Geosat and the ERS-1 (12-km spatial resolution) in 1997 [9] to a global 500-m bathymetry map utilizing Cryosat-2 and Jason-1 data in 2014 [10]. Analytical, semi-analytical and empirical methods have been developed for the estimation of bathymetry up to 30-m depth (Table 1 at reference [1]). The analytical and semi-analytical methods are based on the physics of light transmission in water using different parameters of the atmospheric, water surface, water column, and bottom layers; such parameterization renders these methods more complicated and of greater computational demand to retrieve bathymetry data, but also of higher accuracy than the empirical methods [1]. Lyzenga showed that bathymetry can be estimated over clear shallow water using satellite remote sensing data with a multi-band log linear algorithm. Since then, this method has been utilized in various approaches or with small modifications to derive bathymetry using different spaceborne data [11–15]. Depending on the application and the scale of data needs, high spatial resolution data have become crucial to characterize seascape morphology at local scales, for use in spatial ecology, maritime spatial planning, and navigation. In the last ten years, the advents in remote sensing technology have given birth to satellites with image acquisitions of higher frequency and lower pixel size, e.g., Landsat 8 (30-m and 16 days, Sentinel-2 (10-m and 5 days). The exemplar of the two latter satellite missions—owing to their open, free, and public data access policy—has allowed new scientific developments and operational applications in coastal SDB. In parallel, the Digital Globe's commercial constellation of WorldView and Quickbird satellites has been also offering sub-meter spatial resolution and revisit times of a single day, yet at a high and elusive cost for many institutions.

The majority of the available satellite platforms provide remotely sensed data at an either infrequent temporal resolution or expensive data provision. This gap starts to be filled in 2013 by a new company bridging the gap between the high spatial and high temporal resolution of satellite remote sensing data. Planet Labs, Inc. (http://planet.com) has successfully built and launched 281 CubeSats since 2013 at various phases. Now (2019), it has more than 148 satellites in sun-synchronous orbit which image nearly all off the global land surface and coastal marine surfaces at 3–5-m resolution daily. As such they provide near real-time imagery to the private industry, academic domain, and governmental organizations. The satellites are the so-called CubeSats 3U—about the size of a wine box (10 ⊣ 10 ⊣ 30 cm) carrying a four-band multispectral camera and power/downlinking equipment. Having small size and being built at lower costs, the CubeSats have the potential to overcome tradeoffς among high spatial and temporal resolution because of the multi-satellite constellation approach. This is linked to the mass production of the hardware and low launch costs using various platforms, driving to affordable solutions for commercial satellite companies as well as non-profit and research institutes. A drawback related to the image quality is that the multispectral imagery is acquired using inexpensive sensors at different batch productions with variable radiometric quality, consistency, and signal-to-noise ratio in comparison to the space agency-funded missions (Landsat and Sentinel series) and the commercial platforms (e.g., Marxan Technologies and WorldView satellites) [16]. So far, the CubeSat have limited use in the natural environment mainly due to image quality related to the user needs and the among satellites cross-sensor calibration approaches. In the seascape community, even if the constellation has great potential in transforming coastal remote sensing, few studies have come out so far [17,18].

The objective of the current study is the first utilization of CubeSat imagery to calculate Satellite-derived Bathymetry for a selected site in Crete, Greece, using a plethora of single images from

the same month. Implementing the selected images, we apply the empirical method by Lyzenga [9], which requires only in-situ depth soundings. Based on the best fitted training model, we proceed by applying low pass filters for the enhancement of radiometric anomalies at neighboring pixels, and the calculation of bathymetry for the depth zones according to the International Hydrographic Organization (IHO). The latter approach provides insights into the suitability of the CubeSats for spaceborne bathymetry and of the results to the requirements of the IHO for the inclusion of such products in the production line of navigation charts.

## 2. Materials and Methods

### 2.1. Study Site and Insitu Data

Obros Gyalos is a small protected cove (Figure 1) located at the prefecture of Chania in the island of Crete; it lacks a proximity to rivers and anthropogenic activities. The cove has been selected for the creation of a SCUBA diving park of Crete due to its unique seascape morphology. There was a detailed bathymetric survey among the activities for its establishment during summer of 2017 [19]. In total, 9954 bathymetric points have been collected; these have been split into two parts for calibration and validation. Prior to the random spatial split using the "*Subset features*" tool in ArcGIS 10.5, which randomly splits the dataset into two parts based on percentages, an aggregation of the in-situ data has been performed to match the spatial resolution of the satellite data (3-m). Points have been converted to 3-m pixels using the MEAN function (the mean of the attributes of all the points within the cell) of the corresponding tool "*Point to Raster*" in ArcGIS 10.5. After that, the resulting raster dataset has been converted into points, producing the final dataset of 4756 points, split into 2854 points for training and 1902 points for validation.

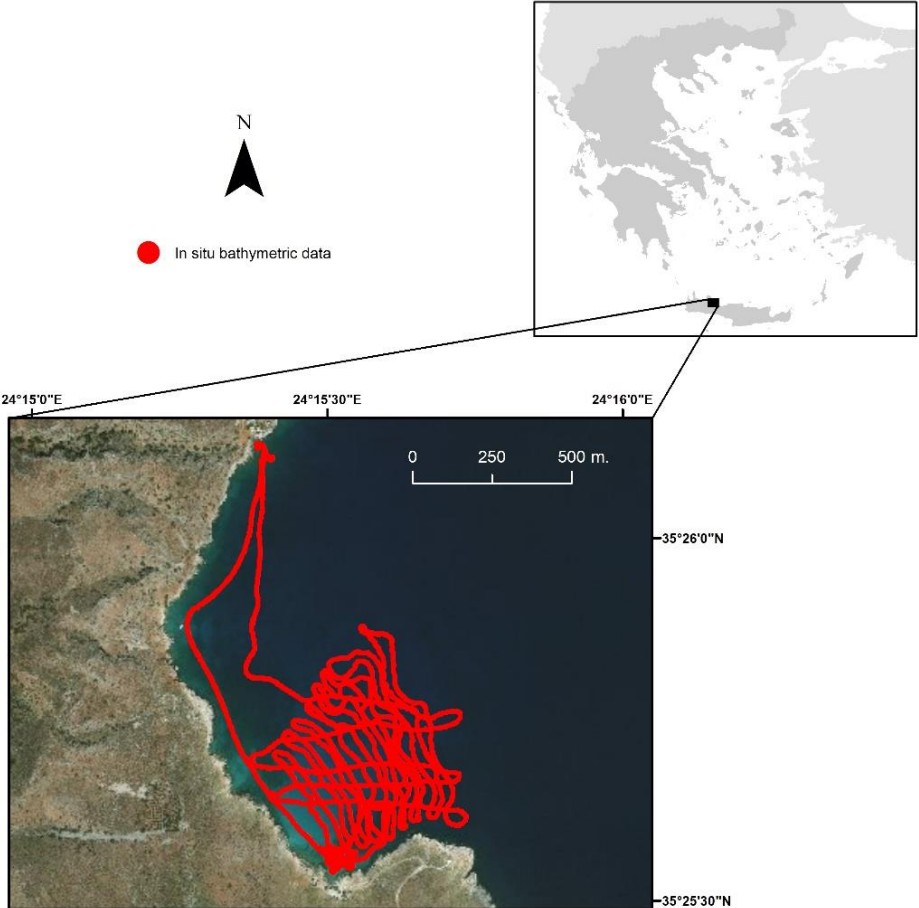

**Figure 1.** Obros Gyalos study site at Apokoronas area; with red dots the collected in-situ data using a single beam echosounder.

## 2.2. Satellite Remote Sensing Data

Five PlanetScope images at the level 3B were used in the current research (Figure 2, Table 1). Before the application of the empirical bathymetric models, all the images had been cropped to the defined study area. Images have been obtained for free under the Planet Education and Research program [20]. PlanetScope level 3B image is an orthorectified Ortho Scene Product, while the pixel value is scaled to Top-of-Atmosphere (TOA) radiance (at-sensor); a surface reflectance (SR) product is also available, reducing the need for atmospherically correcting the orthorectified PlanetScope images, projected to a Universal Transverse Mercator (UTM) cartographic projection [21]. The use of five PlanetScope images was necessary as a control for variations in the radiometric quality of PlanetScope images, and in the atmospheric and water surface conditions (sunglint, wave formation). While signal-to-noise ratio defines the quality of an image band, the current distribution of the PlanetScope CubeSats lacks this information in the metadata, thus we are not able to evaluate the quality of each selected image based on this metric. A multi-image assessment approach limits the chance factor in the conclusions regarding the performance of PlanetScope images—due to the radiometric quality and the atmospheric/water surface conditions of a single image. From the satellite images, the spectral bands blue (455–515 nm), green (500–590 nm), and red (590–670 nm) have been used for the regressions, while the near-infrared (780–860 nm) has been used for masking the land.

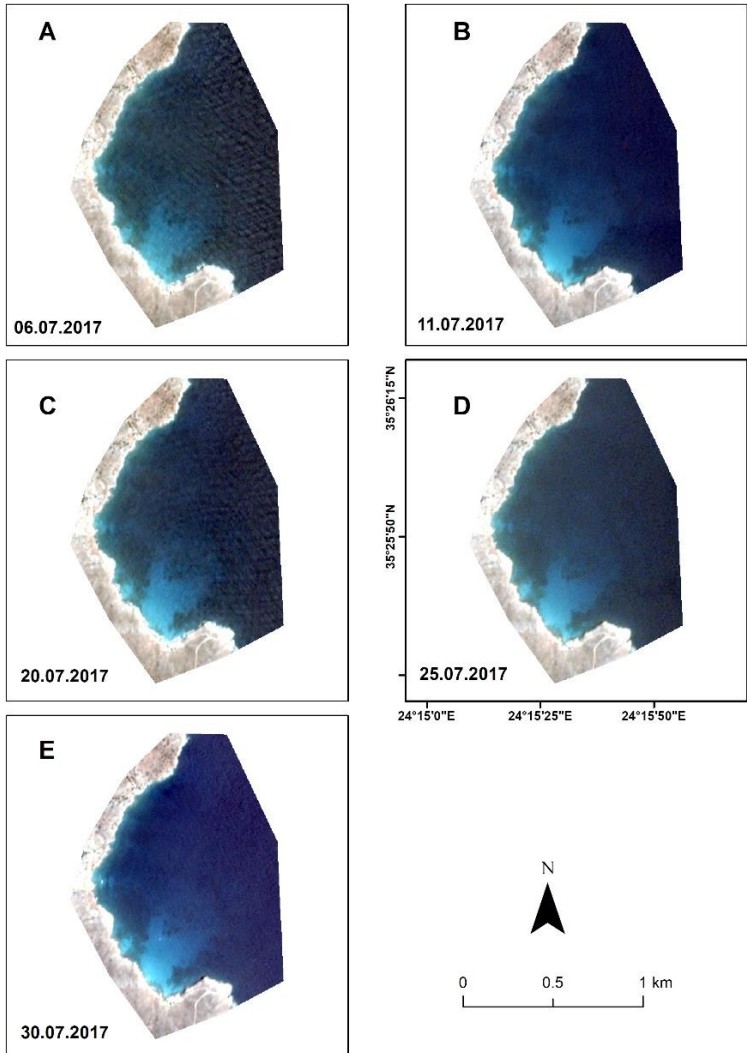

**Figure 2.** The selected daily images of the PlanetScope. All panels (**A–E**) share scale and north arrow. All maps share the same grid as of panel **D**.

**Table 1.** The PlanetScope CubeSat images used in the analysis.

| Scene ID | Date | Time (UTC) | Sun Azimuth | Sun Elevation |
| --- | --- | --- | --- | --- |
| 20170706_082033_103e_3B_AnalyticMS_SR | 6 July 2017 | 8:20 | 106.24 | 59.69 |
| 20170711_082041_1012_3B_AnalyticMS_SR | 11 July 2017 | 8:20 | 107.14 | 59.27 |
| 20170720_082118_101d_3B_AnalyticMS_SR | 20 July 2017 | 8:21 | 109.4 | 58.37 |
| 20170725_082251_1011_3B_AnalyticMS_SR | 25 July 2017 | 8:22 | 111.02 | 57.78 |
| 20170730_082318_102e_3B_AnalyticMS_SR | 30 July 2017 | 8:23 | 112.69 | 57.17 |

### 2.3. Empirical Satellite-Derived Bathymetry (SDB)

Satellite-derived Bathymetry started during the 1970s with empirical methods that used spectral bands at the visible wavelength; blue, green, and red have been widely used as the independent variables in a multiple regression approach, and the in-situ data, the depth soundings with known depths, as the dependent variables. Lyzenga [9] (hereafter Lyzenga85) was the first that developed the equation of the estimation of bathymetry through the aforementioned approach assuming that the relationship between the log-transformed bands and known depth via multiple regression is linear. The coefficients of the regression are applied to the satellite data for the calculation of the bathymetry [19]. For the five images, multiple linear regressions, using the package "*car*" of R [22], have been performed for the calculation of the coefficients. Based on the lower value of the corrected

Akaike Information Criterion (AICc) [23,24] calculated by using the R package "*AICcmodavg*" [25], the selected image has been further analyzed by applying a low pass filter of $3 \times 3$ to reduce the potential radiometric anomalies between pixels. Two regressions have been applied using the training data; the first for the depth zone between 0–10 m and the second for the zone between 10–25 m; the respected coefficients have been applied to the selected image and for each depth zone the metrics "coefficient of determination ($R^2$)", "Standard Error (SEz)" and the "Root Mean Square Error (RMSEz)" have been estimated using the validation points. The comparison of the RMSEz value with the Category Zone of Confidence (CATZOC) values provides insights into the Zones of Confidence and the reliability of the bathymetry product for implementation in navigation charts produced by hydrographic offices [15].

## 3. Results

The availability of the full archive of the PlanetScope imagery allows us the selection of suitable images within the same month, setting two criteria: The absence of sunglint which poses an extra processing step and eventually could introduce additional noise to the resulting deglinted images; this could in turn reduce the suitability of the images due to the already known low signal-to-noise ratio [18]; and the cloud free scenes avoiding cloud masking. Thus, the only difference between the selected images is a slightly visible wavy water surface caused by local winds formed during the morning of the day of acquisition. No sedimentation in the water column was observed, while the bottom cover is mainly composed by two types, bright sandy bottom, and rocky formations (Figure 3). Images have been selected with approximately five days interval. By applying the method described in 2.3 using the subset of training in-situ data ($n = 2854$) and based on the AICc values (Table 2), the image of 11 July 2017 has been selected for further analysis.

The multiple regression results of the training, based on the coefficient of determination and the standard error on the RAW (Surface reflectance) and the transformed ($3 \times 3$ median low pass filter) results are presented in Table 3, while the validation results in Figures 4 and 5, and the produced bathymetric map after the combination of the two different results from the two predefined bathymetric zones in Figure 6.

**Table 2.** Comparison of AICc values calculated by the multiple linear regressions for the SDB estimation of the in-situ data (training) and the log-transformed spectral bands (Blue-Green-Red) of the PlanetScope imagery at full depth range (0–25 m), (Modnames = Model names, AICc = Corrected AIC, Delta_AICc = delta AIC, AICcWt = weight of AICc, LL = Log-liklehood).

| ID | Modnames | AICc | Delta_AICc | AICcWt | LL | $R^2$ |
|----|----------|------|-----------|--------|-----|-----|
| 1 | sdb.glm0706 | 12497.5 | 1128.5 | $8.584 \times 10^{-246}$ | −6243.7 | 0.84 |
| 2 | sdb.glm0711 | 11368.9 | 0 | 1 | −5679.4 | 0.89 |
| 4 | sdb.glm0720 | 12143.5 | 774.6 | $6.203 \times 10^{-169}$ | −6066.7 | 0.86 |
| 5 | sdb.glm0725 | 12989.2 | 1620.3 | 0 | −6489.6 | 0.81 |
| 6 | sdb.glm0730 | 12583.9 | 1214.9 | $1.51 \times 10^{-264}$ | −6286.9 | 0.84 |

**Table 3.** The results from the multiple regressions using the training data on the RGB spectral bands of the image of 11 July 2017 in the depth zones 0–10 m and 10–25 m.

| Regression Statistics | RAW (0–10 m) | $3 \times 3$ (0–10 m) | RAW (10–24 m) | $3 \times 3$ (10–24 m) |
|-----------------------|--------------|----------------------|---------------|------------------------|
| Multiple R | 0.93 | 0.94 | 0.9 | 0.92 |
| R Square | 0.86 | 0.88 | 0.81 | 0.84 |
| Adjusted R Square | 0.86 | 0.88 | 0.81 | 0.84 |
| Standard Error | 0.72 | 0.66 | 1.63 | 1.48 |
| Observations | 702 | 702 | 2152 | 2152 |

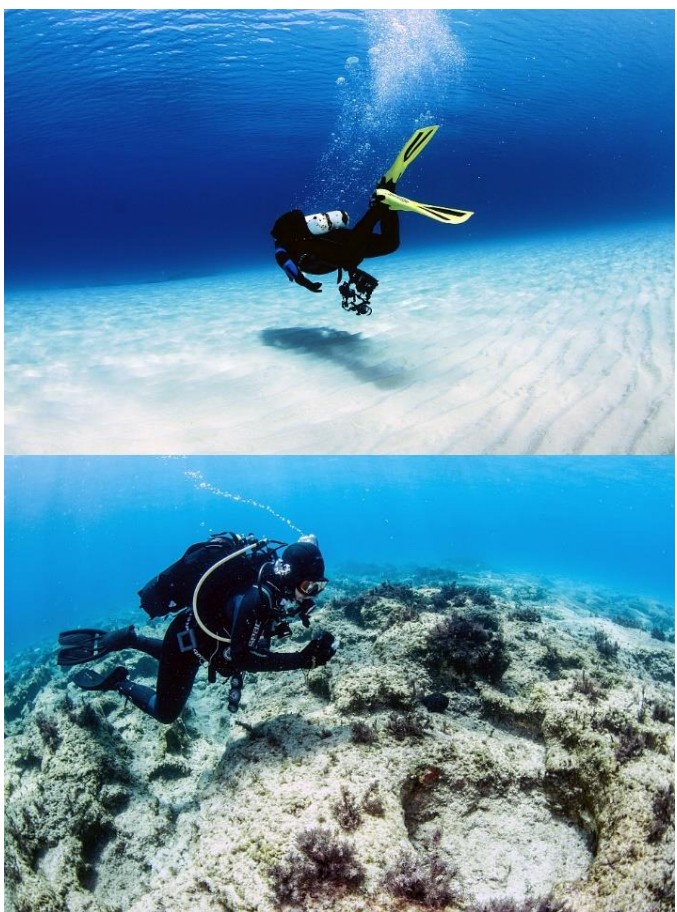

**Figure 3.** The two seabed cover types in the project area; above: sandy soft bottom, below: carbonated rocky surfaces partially covered by brown macroalgae.

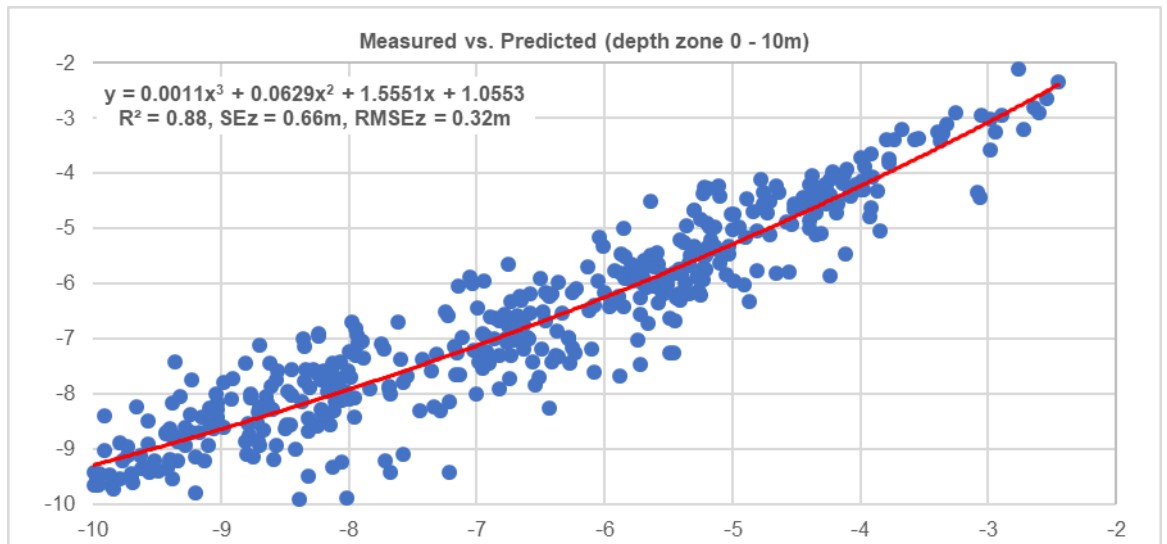

**Figure 4.** The validation plot for the depth zone of 0–10 m of in-situ depth points (*x*-axis) against image-derived bathymetries (*y*-axis) implementing the Lyzenga85 model on the image of 11 July 2017. 3rd order polynomial equation has been applied.

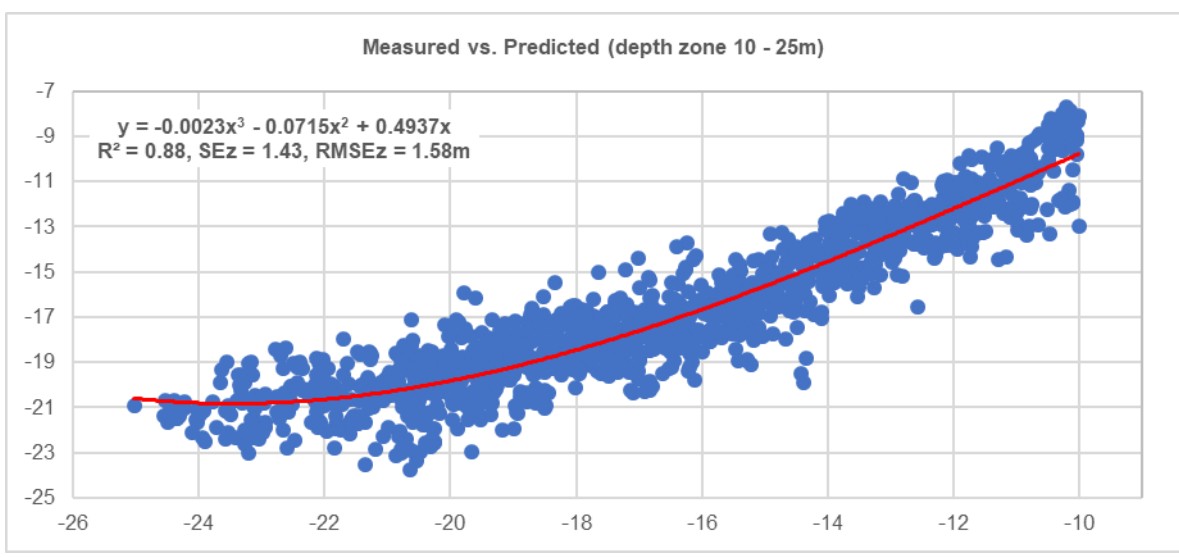

**Figure 5.** The validation plot for the depth zone of 10–25 m of in-situ depth points (*x*-axis) against image-derived bathymetries (*y*-axis) implementing the Lyzenga85 model on the image of 11 July 2017. 3rd order polynomial equation has been applied.

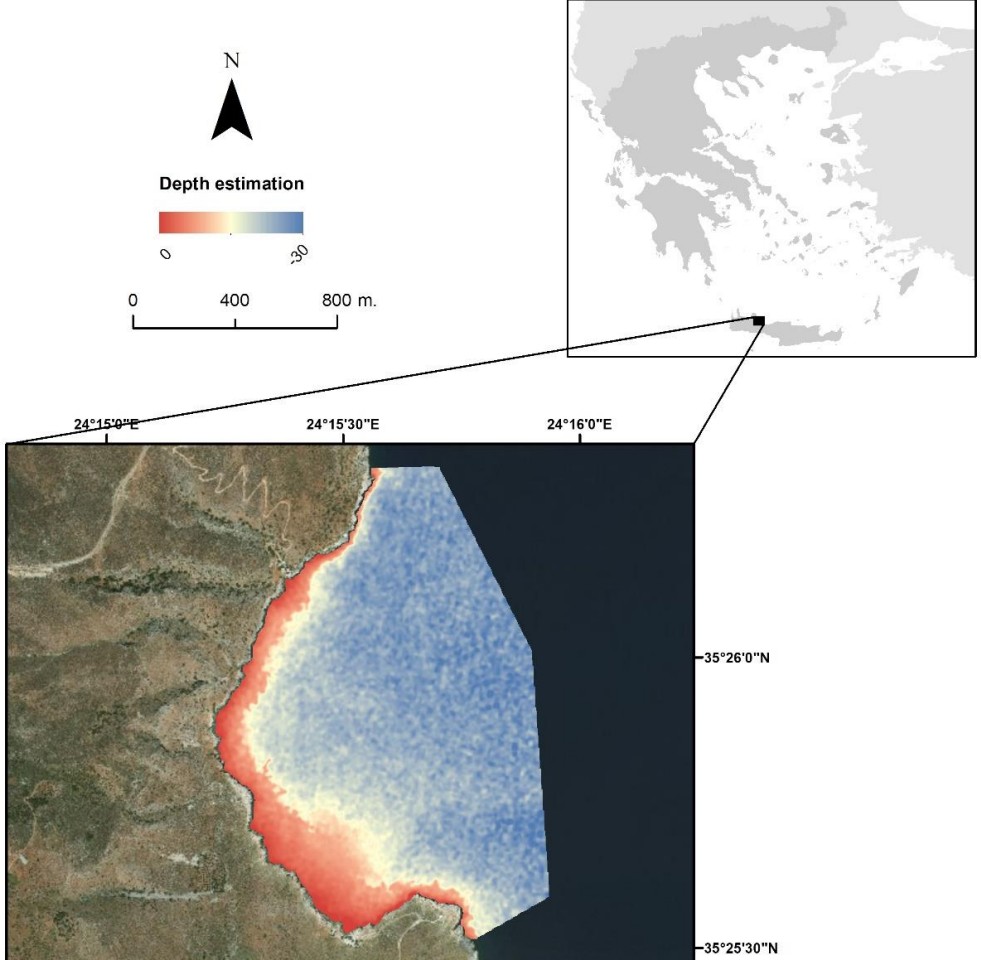

**Figure 6.** The Satellite-derived bathymetry estimated by the combination of the two-depth zones on the image of 11 July 2017.

## 4. Discussion

### 4.1. The CubeSats and the Performance of the Models

PlanetScope constellation is a unique Earth Observation fleet with daily revisit and almost 3-m pixel size, providing new insights into the observation and monitoring of nearly any place on Earth; this allows for new endeavors in mapping, change detection, and monitoring of both land and coastal zone areas. It is worth mentioning that this constellation has been based on CubeSats; they have been built using inexpensive electronics meaning that they are not direct inter-comparable as regards to the radiometric quality, consistency and signal-to-noise ratios of the commercial and agency-funded satellites [26]. Data from the fleet have been already implemented for tracking vegetation dynamics, hydrological applications, Digital Elevation Model creation, seabed cover and coral reef mapping [17,18,27–29]. The results from the recent studies are promising, while global projects like the Allen Coral Atlas—aiming at mapping the swallow (<15 m) global coral reefs using solely PlanetScope imagery [30]—show the importance of the availability of high spatial and temporal resolution satellite data for mapping and monitoring areas where persistent cloud cover is a barrier either for the open access data from USGS Landsat and Copernicus Sentinel-2 or for the data coming from the commercial satellite platforms; the latter have higher spatial resolution (<2 m) and better signal-to-noise ratio, but the data flow is in sparse intervals, making a cloud or almost cloud-free image a challenge.

The limitations of the present study are method and technology-wise. First, our implemented methodology incorporates an in-situ bathymetric dataset of 4726 points to both train and validate the satellite-derived bathymetry model. This violates Tobler's first law of geography—geographically neighboring observations have a tendency to be alike the ones further apart—introducing a statistical bias in our approach. Nevertheless, spatially independent field-based calibration and testing datasets are sparse due to reasons of practical nature, i.e., the high acquisition cost of such datasets in situ. Technology-wise, there are two main limitations that we could generalize for the majority of the coastal aquatic scientific and operational developments and applications with Planet's CubeSats: (a) Radiometric differences between the different PlanetScope CubeSats; (b) low signal-to-noise ratio for high-accuracy detection and mapping of the coastal benthos. We expect the present and near-future deployments of new Planet satellite sensors featuring higher radiometric quality to overcome the aforementioned technological limitations and improve the accuracy and efficiency of similar approaches. The present study is the first one that tries to understand the performance of PlanetScope CubeSats in the estimation of bathymetry in a mixed-bottom seascape using available in-situ data—collected for different purposes; this is carried out by applying empirical-based statistical relationships between the log-transformed, water-penetrating bands (most frequently at blue and green wavelengths but also including the red wavelength) and acoustic-derived in-situ depth data.

Having access to the full archive of the imagery, we selected five images over July 2017, to account for the effect of sea surface conditions that change due to waves and also for the radiometric consistency of the different flocks, as the CubeSats have been released in varying time frameworks since 2013. From the application of the Lyzenga85 empirical method in the five images and based on the AICc values from the training models (Table 2), we selected and further employed the image with the lower AICc value and the best $R^2$ (Figure 2B). The visual examination of the selected image (Figure 2B) lacks a wavy surface and visible noise in comparison to the other images that can possibly be caused by high altitude haze and the older electronics that the first flocks have been made of. The application of a low pass filter ($3 \times 3$ median) to the surface reflectance imagery improves the results of the training of the method (Table 3) as it smooths possible pixel anomalies. The bathymetric zonation of the multiple regressions into two groups (0–10 m and 10–25 m) has been selected to understand the accuracies of the SDB in relation to the requirements of the IHO in the inclusion of bathymetry data in navigation maps and the corresponding ZOC that the depth accuracies fall into [3]. By splitting the analysis into

these two zones, coefficients from the multiple regressions are applied into the respective bathymetric zones, while the final map is the unification of the results from the two zones.

*4.2. Are PlanetScope Suitable for Data Inclusion in Navigation Maps?*

The results from the application of the calculated coefficients are very promising for the value of the PlanetScope imagery in estimating bathymetry and its subsequent integration in navigation charts compliant with the IHO requirements and the fit of them into ZOCs. The results are also comparable with the Copernicus Sentinel-2 based SDB from the same site, as has been shown by [16] using the Google Earth Engine platform. For the Zone of 0–10 m (Figure 4), the estimated $R^2$, using a 3rd order polynomial equation, is 0.88, the SEz is 0.66 m, and the estimated RMSEz is 0.32 m while for the zone 10 m–25 m $R^2$ is 0.88 with SEz at 1.43, and RMSEz at 1.58 m. According to the IHO, the Zone of Confidence A1 requires data in the zone 0–10 m with a depth accuracy of ± 0.6 m and a position accuracy of ±5 m, while the zone A2 a depth accuracy of ±0.6 m and a position accuracy of ±20 m. For Zone B, it requires for the same depth as in A2, an accuracy of ±1.2 m and a position accuracy of ±50 m. For the depth zone 10–30 m, the ZOC A1 requires ±0.8 m, the ZOC A2 requires ±1.2 m and the ZOC B requires ±1.6 m. The results from the current study suggest that the PlanetScope data can fit into the ZOC A2 and possibly into the ZOC A1. This fit is also supported by the position accuracy of the satellite images, according to the recent study of [31], which shows that the geolocation performance of the PlanetScope's Level 3A product is good and the absolute geolocation performance is set by a max $RMSE_x$ = 5.18 m, $RMSE_y$ = 4.21 m and CE(90) = 9.93 m, respectively.

While the current study includes only one site and is targeted at understanding the behavior of the different selected PlanetScope CubeSat images from the same site covering one-month period, more studies are required to robustly conclude for the suitability of the employed Earth Observation data in the calculation of bathymetry. Studies that will include different satellites of very high-resolution data and high signal-to-noise ratio with images from the same period (e.g., within the same month) will allow a full benchmark on the radiometric quality of the PlanetScope data for the calculation of SDB. Other approaches like the cloud-based one presented in [19] will allow the full scientific and operational exploitation of the Planet archive for seabed mapping and monitoring without the laborious analysis per image and the visual inspection of the "best image" for further processing and analysis; this will be achieved through the utilization of multi-temporal (weekly, monthly, seasonal) PlanetScope image composites, which will reduce technological and environmental issues, both intra and inter-sensor, e.g., low signal-to-noise ratio, varying radiometric quality, atmospheric, water surface and column, and seabed conditions.

There is an increasing need in the charting of shallow coastal waters across the globe at fine scales that will allow the creation of new navigation maps, but also the update of old productions. IHO has started to adopt the SDB approach and the IHO S-44 standards are currently under revision. The revised one will be updated to include the ability to exploit new technologies for the update of nautical charts when that is possible under the specifications required, as the use of traditional means, like the hydroacoustics, in shallow uncharted waters is dangerous for the equipment and the per km$^2$ charted area is much more costly in comparison to the SDB approach.

## 5. Conclusions

The present study is the first one that attempts to examine the performance of the PlanetScope CubeSats in calculating Satellite-derived Bathymetry and explores whether the results fit into the requirements of IHO for nautical maps of navigation. The availability of almost daily satellite images in the archive allows the selection of the most suitable data based on cloud coverage, water surface conditions, and intense visible sunglint—an asset of the high spatial and temporal satellite constellation. The results from the applied empirical method with the intermediate preprocessing steps are promising and show that it has great potential for coastal bathymetry estimation, especially in the shallow waters. However, given that one site has been tested, more work is needed to understand the nature of

PlanetScope CubeSats in estimating SDB by including several sites distributed in different water bodies that cover both case I and II waters and systematic collection of in-situ soundings that correspond to each month. Also, the analysis of monthly and annual image composites, as these are provided by Planet for commercial use, will support the elimination of issues related to the absence of cloud mask information, and the low signal-to-noise ratios of the PlanetScope imagery. All in all, given that the technology of CubeSats is improved (i.e., higher signal-to-noise ratios and more spectral bands in the visible wavelengths), we expect, in the next decade, a boom of fleets that can be eventually exploited to carry out scientific and operational mapping and monitoring of the coastal aquatic environment at a fraction of the cost of "traditional" satellite platforms.

**Author Contributions:** D.P. conceived the idea, performed the analysis, and prepared the manuscript; D.T. supported the analysis and contributed to the manuscript. N.C. and P.R. contributed to the manuscript.

**Funding:** D.P. and N.C. are supported by the European H2020 Project 641762 ECOPOTENTIAL: Improving future ecosystem benefits through Earth Observations. D.T. is supported by a DLR-DAAD Research Fellowship (No. 57186656).

**Acknowledgments:** This work contributed to and was partially supported by the European H2020 Project 641762 ECOPOTENTIAL: Improving future ecosystem benefits through Earth Observations. Dimosthenis Traganos is supported by a DLR-DAAD Research Fellowship (No. 57186656). We would like to thank the two anonymous reviewers for their constructive comments that improve our work. We also would like to thank Planet's Ambassadors program as Planet satellite images (Planet Team, 2017) are provided through this program.

**Conflicts of Interest:** The authors declare no conflict of interest.

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
