# Peer review of "Cubesats Allow High Spatiotemporal Estimates of Satellite-Derived Bathymetry"

_remotesensing, doi:10.3390/rs11111299_

Round 1

Reviewer 1 Report

The present study examined the performance of the PlanetScope CubeSats in calculating Satellite-derived Bathymetry. 

It is generally interesting to me, and can be accepted for publication after major revisions.

Major comments:

The availability of almost daily satellite images in the archive allows the selection of the most suitable data based on cloud coverage, water surface 

conditions and intense visible sunglint―an asset of the high spatial and temporal satellite constellation. 

The results from the applied empirical method with the intermediate preprocessing steps are promising 

and show that it has great potential for coastal bathymetry estimation, especially 

in the shallow waters. 

However, given that one site has been tested, more work is needed to understand the nature of PlanetScope CubeSats 

in estimating SDB by including various locations in different water bodies that cover both case I and II waters 

and systematic collection of in situ soundings that correspond to each month. 

(1) Please add tests in different seasons if possible.

(2) Please add validation at another site if possible.

(3) Please add detailed discussions on the possible uncertainty that has not been tested or accounted for.

minor comments:

1. please indicate the time periods at the figure captions for the results in Figs. 4, 5, 6.

2. The quality of Figs. 4 and 5 can be further improved.

3. Figure 6:   A multi-color legend is recommended to clearly illustrate the different depth estimations.

Author Response

Dear reviewer, in the pdf you will find our responses to the useful comments.

Reviewer 2 Report

This study was conducted with the great advantage of having the extensive field sampled bathymetric data and the PlanetScope images that were available for free.

While this is an interesting and important study as described by the authors, the following could be considered to improve the presentation and quality of the study for a peer-reviewed publication.

The maps showing the study area (Fig 1 for locations for in situ data) should be presented at a larger map scale. The color scale for depth in Figure 6 needs to include multi-color (e.g. blue-red) for better visualization and descretion. Perhaps, an in-situ data driven bathymetric map can be presented together with the same color scale with the current map of Fig. 6.

Due to the excellent water clarity condition of the area, we easily expect to see good depth estimation results using multi-band visible light satellite images, not only from the high spatial and high temporal resolution CubeSats but from the conventional broad band satellite images. I suggest that the authors apply the similar approach (the same depth estimation algorithm) to other satellite data available in the same month/year of the area. And assess their accuracy with the in situ data (like Figs. 4 and 5) in order to compare with the performance of the present study results. 

Author Response

Dear Reviewer, in the attached pdf you will find the response to the useful comments

Round 2

Reviewer 1 Report

The authors have not well addressed my comments, although they have difficulty to add further validations. At least, the figure quality should be improved largely to meet the level of this journal.  I would leave the decision to editor if this manuscript is suitable for Remote Sensing or not.

Author Response

Dear Reviewer,

Unfortunately, we don't have similar dataset from other site to include in our analysis.

The figures are provided in 300dpi as TIFF files - Assume that the final version will show them as they are at high quality.

Dimitris Poursanidis

Reviewer 2 Report

Please see my comments in the attachment.

Author Response

Dear Reviewer,

We would like to thank you for the comments on the manuscript.

All mentioned points have been addressed.

Dimitris Poursanidis

Round 3

Reviewer 1 Report

I have no further comments on the scientific contents. But, I still believe that the figures (Figure 1, 5, and 6) in this paper can be further improved/refined, e.g., figure layout, lables and font size, etc.. The figure resolution is not the major problem in the presentation.

Author Response

Dear Reviewer,

The figures (apart from the figures of the plots) have been uploaded in a zip file at 400 dpi for use in the production as seems that the word document compress them and cause issues. We have rework the maps to be more readable.

Hope that now it can be accepted for publication.

Looking forward

Dimitris